# MFFAMM: A Small Object Detection with Multi-Scale Feature Fusion and Attention Mechanism Module

**Zhong Qu** [1,*], **Tongqiang Han** [1] **and Tuming Yi** [2]

1   College of Computer Science and Technology, Chongqing University of Posts and Telecommunications, Chongqing 400065, China
2   Institute of Information Technology Southwest Computer Co., Ltd., Chongqing 400065, China
*   Correspondence: quzhong@cqupt.edu.cn

**Abstract:** Aiming at the low detection accuracy and poor positioning for small objects of single-stage object detection algorithms, we improve the backbone network of SSD (Single Shot MultiBox Detector) and present an improved SSD model based on multi-scale feature fusion and attention mechanism module in this paper. Firstly, we enhance the feature extraction ability of the shallow network through the feature fusion method that is beneficial to small object recognition. Secondly, the RFB (Receptive Field block) is used to expand the object's receptive field and extract richer semantic information. After feature fusion, the attention mechanism module is added to enhance the feature information of important objects and suppress irrelevant other information. The experimental results show that our algorithm achieves 80.7% and 51.8% *mAP* on the PASCAL VOC 2007 classic dataset and MS COCO 2017 dataset, which are 3.2% and 10.6% higher than the original SSD algorithm. Our algorithm greatly improves the accuracy of object detection and meets the requirements of real-time.

**Keywords:** small object detection; multi-scale; feature fusion; attention mechanism; receptive field





## 1. Introduction

Object detection is a research topic in digital image processing, which mainly includes two sub-tasks of object positioning and object classification. It is widely used in intelligence monitoring [1], autonomous driving [2], industrial detection [3,4], remote sensing image analysis [5], aerospace detection [6], and other fields. Nowadays, with the rapid rise of deep learning, new research ideas and directions have been brought to object detection. Due to the variability, complexity, and morphological heterogeneity of object detection, problems such as the detection accuracy of small objects are low [7–9]. To improve the accuracy of small object detection, we propose a method based on a multi-scale feature fusion and attention mechanism module.

At present, deep learning-based object detection algorithms are developing rapidly, which can be roughly divided into two-stage object detection algorithms and single-stage object detection algorithms. The core of the two-stage algorithm is to replace the artificial feature extraction method with a convolutional neural network. Girshick et al. proposed a deep learning-based object detector R-CNN (Region Convolutional Neural Network) [10]. However, due to a large amount of calculation of the model, the detection efficiency is low. He et al. proposed the SPP-Net (Spatial Pyramid Pooling Network) [11] algorithm. Girshick et al. proposed the Fast R-CNN [12] to introduce the mapping relationship between the image and the feature layer. Ren et al. proposed the Faster R-CNN [13]. Based on Fast R-CNN, RPN (Region Proposal Network) was added, which can improve the network calculation speed. Since the two-stage detection method needs to extract candidate regions, and then make secondary corrections based on the candidate regions to obtain the detection results, the feature load of each layer is too large. It is difficult to achieve real-time detection and high speed.

To solve the speed problem of the two-stage object detector, Redmon et al. proposed YOLO (You Only Look Once) [14], but the positioning and classification accuracy of the algorithm was relatively low. Liu et al. proposed the SSD (Single Shot MultiBox Detector) [15], which largely improves the detection speed and detection precision of the single-stage object detector. Fu et al. proposed the DSSD (Deconvolutional Single Shot Detector) [16], which used the Residual-101 Network [17] as the backbone with a stronger feature extraction capability. Li et al. proposed the FSSD (Feature Fusion SSD) [18]. Redmon and Farhadi proposed YOLOv2 [19], which improves YOLO's speed and precision. Redmon and Farhadi proposed YOLOv3 [20], which used a newly designed Darknet-53 residual network combined with FPN (Feature Pyramid Network) [21] for multi-scale feature maps fusion, and the performance of the object detector was further improved, but the accuracy of identifying the object position is poor, and the use of a fixed bounding box in each grid reduces the number of candidate frames.

Given the deficiencies of the above models, we propose an object detection method based on a multi-scale feature fusion and attention mechanism module. Our main contributions are as follows:

(1) Based on the SSD model, we deconvolute and amplify the deep feature layer, then fuse it with the shallow feature layer, fully combining the detailed information in the shallow network and the semantic information in the deep network to improve the accuracy of small object detection.

(2) We add a channel attention mechanism to the network, and introduce a GC (Global Context) block as an attention module between different scales of the network, so that the network has the ability of global context modeling and the advantage of remote dependency capture, retains more object's features information, suppress irrelevant other information and improve the ability of the network to extract features, thereby enhance the detection accuracy.

(3) At the same time, we add the RFB (Receptive Field block) module to expand the receptive field of the object feature in the original image. By simulating the human visual receptive field, the feature receptive field is expanded to obtain more objective information.

The rest parts of this paper are arranged as follows: We give a brief introduction to the research related to this paper in Section 2. Section 3 introduces the structure and training process of the new template network based on SSD. Section 4 displays the experimental results and proves the superiority of our proposed network. Section 5 gives the discussion and conclusion.

## 2. Related Work

### 2.1. SSD

The SSD algorithm balances the detection speed and accuracy of the single-stage object detector. Besides, it also reveals strong robustness to different object sizes. SSD uses VGG16 [22] as the backbone and the new convolutional layer is used to produce more feature maps for detection. The model structure of the SSD is shown in Figure 1. It obtains six different scale feature maps through the VGG network, the sizes of which are $38 \times 38$, $19 \times 19$, $10 \times 10$, $5 \times 5$, $3 \times 3$, and $1 \times 1$, a series of fixed-size default boxes are set on the obtained feature map.

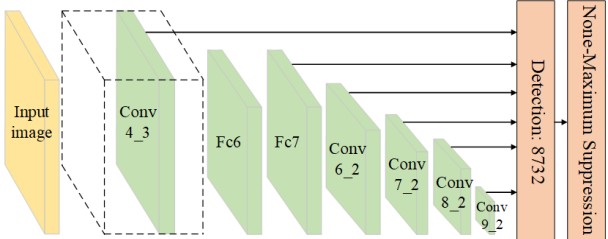

**Figure 1.** The network structure of the conventional SSD. The feature pyramid consists of Conv4_3, Conv7, Conv8_2, Conv9_2, Conv10_2, and Conv11_2. The corresponding feature size is $38 \times 38$, $19 \times 19$, $10 \times 10$, $5 \times 5$, $3 \times 3$, and $1 \times 1$. NMS algorithm is used to eliminate redundant detection boxes.

### 2.2. Attention Mechanism

People's visual system tends to pay attention to the part of the information that assists judgment in the image and ignores irrelevant information. This mechanism can be introduced into the process of the convolutional layer encoding the input information and outputting the feature map, which improves the network's ability to extract and use the feature. Hu et al. proposed SENet (Squeeze-and-Excitation Networks) [23]. Wang et al. proposed NLNet (Non-Local Neural Network) [24] to weigh all pixels according to the correlation between pixels. Cao et al. combined SENet and NLNet and proposed GCNet (Global Context Network) [25]. From the above attention method, the attention mechanism can make the neural network pay more attention to key information, and improve the feature extraction and utilization ability of the object detection model. Fan et al. propose a multi-scale feature fusion module based on the attention mechanism [26], which adopts a CAM (Channel Attention Module) to reweigh feature maps in different channels to further improve the semantic segmentation results.

### 2.3. Multi-Scale Feature Fusion

In the object detection algorithm based on deep learning, the shallow network pays more attention to detailed information, and the higher layer network pays more attention to semantic information. Therefore, the shallow feature layers and the high-level feature layers are fused to obtain new feature layers, which can get smaller object information, and improve the overall detection accuracy. Cai et al. proposed MS-CNN (Multi-Scale Convolutional Neural Network) [27] which used two subnets and combined features of different scales. Li et al. proposed an indoor small target detection method based on multiscale feature fusion [28], and the target detection layer and its adjacent feature layer are fused in the SSD algorithm.

### 2.4. Receptive Field Block

In object detection, the information of the shallow network is not fully utilized, and the size of the object in the original image can be ensured by increasing the visual receptive field, thereby improving the feature extraction ability of the network to obtain more objective information. Liu et al. proposed the Receptive Field block [29], which was added based on the SSD network architecture. It enhanced the network's feature extraction capability by simulating the receptive field of human vision.

## 3. The Improved SSD Networks

### 3.1. Network Architecture

The improved algorithm incorporates the idea of FPN into the structure to improve the detection effect of objects with a small proportion. It combines low-level features and high-level features through the Top-Down structure to fuse multi-scale features, to make full use of the feature information of each scale convolution. SSD network mainly uses the Conv4_3 layer to predict small objects. We fuse Conv4_3 and Conv5_3, and take the fused convolutional layer as the new prediction layer, at the same time add the RFB module after the prediction layer and use dilated convolution to increase the receptive field without increasing the number of parameters to extract richer semantic information. In addition, an attention mechanism is introduced in our network to highlight the foreground regions and reduce the influence of the background.

The overall structure of the network is shown in Figure 2.

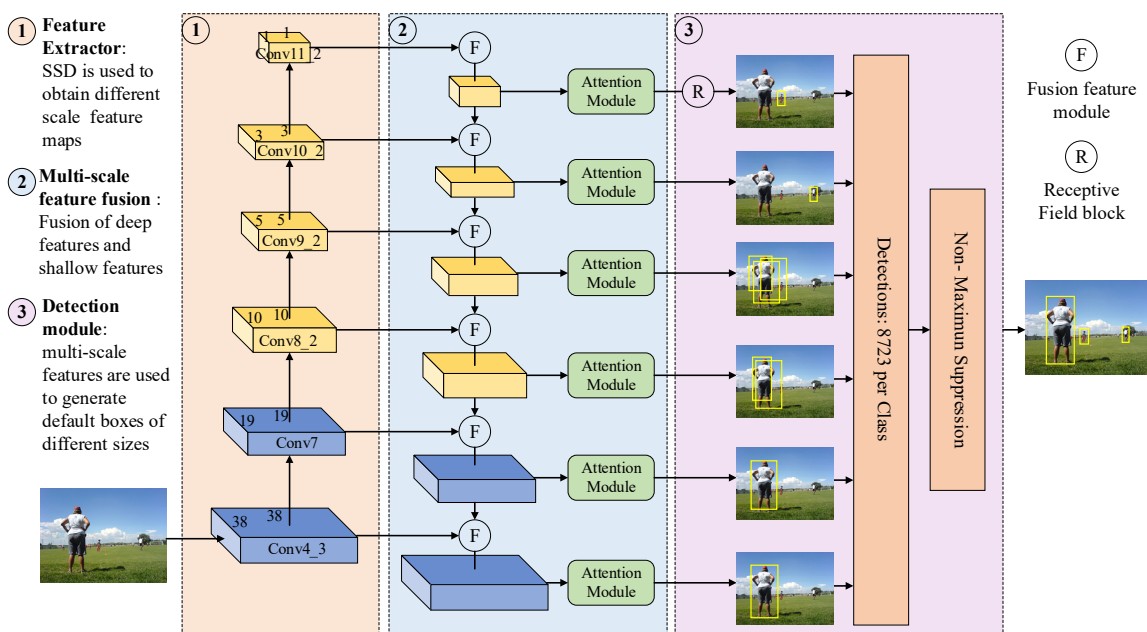

**Figure 2.** The framework of our improved SSD algorithm.

### 3.2. Backbone with Attention Mechanism

To promise the speed of the proposed method, we adopt VGG-16 as our backbone, which is the same as SSD. After the Conv4_3, Conv7, Conv8_2, Conv9_2, Conv10_2, and Conv11_2 layers, we add the global context module in GCNet as the attention module to improve the ability to express semantic information. Figure 3 shows the details of the GC block. It includes three parts, named context modeling module, feature conversion module, and fusion module. The context modeling module first uses a 1 × 1 convolution and softmax function to get the weights of each spatial location and then feeds the weights back to the original image. The feature conversion module performs feature conversion through two 1 × 1 convolutions for dimensionality reduction and dimensionality increase operations. To make the model convergence faster and improve the generalization of our method, a normalization operation is added between the two convolution operations.

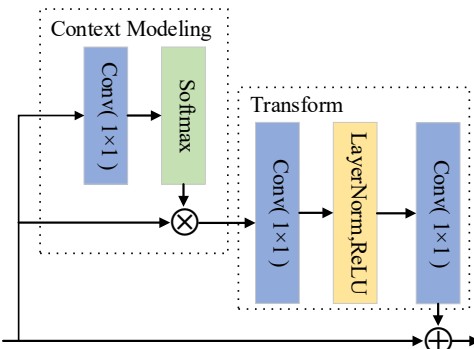

**Figure 3.** The architecture of GC block. This structure consists of three parts: context modeling module, feature conversion module, and fusion module. ⊗ denotes matrix multiplication, ⊕ denotes broadcast elementwise addition.

Finally, the fusion module uses the element addition method to integrate the global context features onto each location.

Equation (1) is defined as follows,

$$z_i = F\left(x_i, \delta\left(\sum_{j=1}^{N_p} \alpha_j x_j\right)\right) \tag{1}$$

where $\sum_j \alpha_j x_j$ represents the context modeling module, which aggregates the features of all positions by the weight $\alpha_j$ to obtain the global context features, $\delta(\cdot)$ represents the feature transformation to capture the dependencies between the channels, $F(\cdot, \cdot)$ represents the fusion module.

### 3.3. Feature Fusion

The SSD algorithm uses the Conv4_3 layer to predict objects with a small proportion, deep convolutions Conv7, Conv8_2, Conv9_2, Conv10_2, and Conv11_2 predict large objects, SSD only uses one shallow convolution to extract the feature information of small objects, which has less semantic information for small objects, resulting in a poor overall detection effect. Refs. [30–32] use shallow feature maps to improve the quality of detection, because shallow features contain more detailed information. The improved algorithm uses feature fusion to merge shallow convolutions and deep convolutions. The outputs of the fusion convolutional layer contain not only more detailed information but also more semantic information. In the SSD network structure, the feature information extracted by the Conv5_3 layer covers more context information than the Conv4_3 layer. Therefore, we fuse the Conv4_3 and Conv5_3 layers. The new fusion layer is shown in Figure 4, Conv5_3 is deconvolved for upsampling so that the size of the extracted feature map is the same as the size of the feature map extracted by the Conv4_3 convolutional layer. The BN layer is added after the two convolutional layers to improve the convergence speed of the network, and the ReLU function is added after the BN layer. The ReLU function simplifies the network model, reduces the calculation, and improves the training speed of the network. By setting many parameter values less than 0 to 1, the generalization ability of the network is indirectly improved. We use the concate fusion method to splice the two feature maps into a 1024-channel feature map. The concate operation directly joins the two original feature maps to the channel, and the process of fusing the features is handed over to the network to learn because it is just simple. Therefore, the feature information of the original feature map is relatively complete.

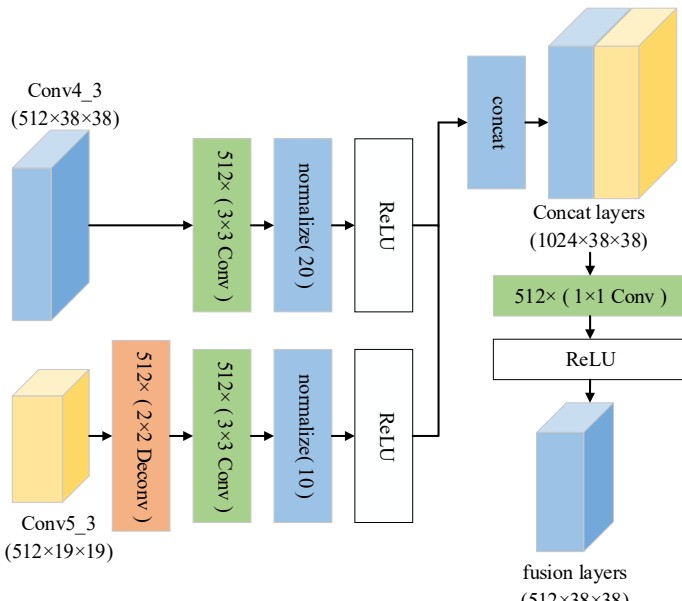

**Figure 4.** Feature fusion module with Conv4_3 and Conv5_3. We use the deconvolution operation to enlarge the high-level feature maps and sum them in a concatenated way. Finally, a $1 \times 1$ convolution is used for dimensionality reduction and feature reorganization.

For other convolutional layers, the improved algorithm adopts the feature fusion method, as shown in Figure 5. Firstly, the original convolutional layer is processed by two $3 \times 3$ convolutional layers to obtain more context information. The small-scale convolution uses deconvolution to keep the size of the large-scale convolution consistent, and then

fuse it by the element-wise method. Finally, we use a $3 \times 3$ convolution to enhance the distinguishability of features. For the last layer of convolution, the method is shown in Figure 5b. There is no upsampling operation, and use three convolutional layers to improve the robustness of the model.

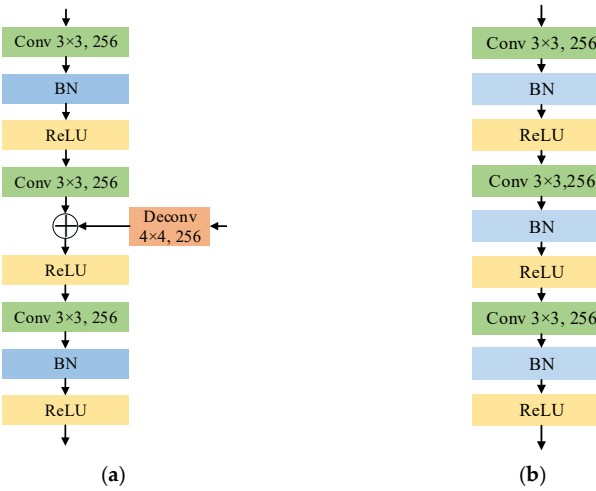

(a)                                    (b)

**Figure 5.** The overview of the feature fusion module. (**a**) Fusion feature module with deconvolution. (**b**) Fusion feature module without deconvolution. $\oplus$ denotes broadcast elementwise addition.

### 3.4. RFB and Dilated Convolution

The structure of the RFB module is similar to the structure of the multi-branch convolution of the inception module, which is composed of multiple branches with different kernels and dilated convolution layers. The multi-branch convolution part uses convolution kernels of $1 \times 1$, $1 \times 3$, $3 \times 1$, and $3 \times 3$. This structure can effectively decrease the number of convolution parameters. The purpose of the dilated convolution part is to generate a higher resolution feature map and increase the receptive field without raising the number of convolution parameters. The dilated convolution adds a hyper-parameter which is called the dilation rate based on standard convolution that refers to the number of kernel intervals. The specific structure is shown in Figure 6. The size of the input image is $300 \times 300$, and the $38 \times 38$ sizes of the feature map are used to predict small objects, but the $38 \times 38$ feature map has less semantic information, which is insufficient to characterize the object information. In this paper, the RFB network structure is added after the $38 \times 38$ prediction layer to extract richer semantic information.

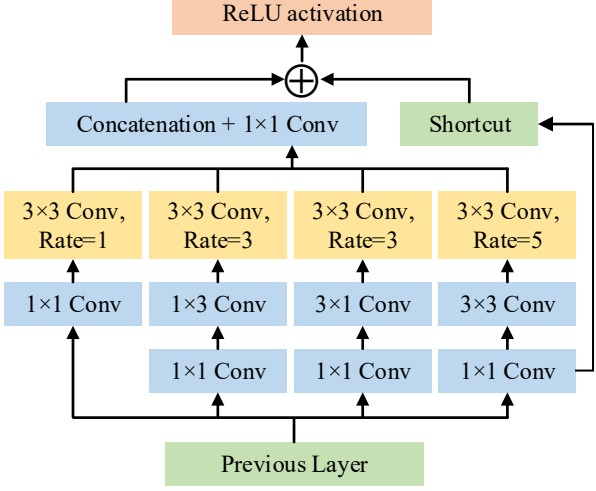

**Figure 6.** The architectures of RFB-s. $\oplus$ denotes concatenate fusion way.

*3.5. Loss Function*

Multiple objects often cluster together in an image, which can easily cause mutual occlusion, and greatly increases the difficulty of object positioning. The detector is easily confused since these objects have similar appearance features. As the detection results are required to be further processed by NMS, the blocked object is easily suppressed by other objects, resulting in missed detection.

Inspired by the literature [33,34], based on SSD's multi-task loss function, we add Repulsion loss to improve the loss function, our overall loss function can be defined as follows,

$$L = L_{SSD} + L_{Rep} \tag{2}$$

where $L_{SSD} = \frac{1}{N}\left(L_{conf}(x,c) + \alpha L_{loc}(x,l,g)\right)$, $L_{conf}$ is the classification loss function, and $L_{loc}$ is the positioning loss function. $N$ is the number of prediction boxes (positive examples) whose prediction box and real box cross ratio is greater than the threshold, and $\alpha$ is the ratio used to adjust the classification and positioning loss functions.

$L_{Rep}$ represents the repulsion loss and repulses other prediction boxes close to this target. $L_{Rep}$ can be defined as follows,

$$L_{Rep} = \sum\nolimits_{p \in P+, g \in G/[g_{iou\_max}]} IOG(p,g) \tag{3}$$

where $P+$ is the set of all positive proposals, and $G$ is the set of all ground-truth boxes in an image. $g$ represents other ground-truth boxes except for the maximum IOU (Intersection over Union), which is the repulsion box corresponding to each prediction box. $IOG(p,g) = \frac{b_p \cap b_g}{b_g}$ represents the ground-truth box with the largest IOU of the prediction box and all ground-truth boxes, then $g_{iou\_max}$ can be calculated as follows,

$$g_{iou\_max} = argmax_{g \in G, p \in P} \frac{b_g \cap b_p}{b_g \cup b_p} \tag{4}$$

It can be seen from Equation (3) that the larger the IOG of the predicted box $p$ and the surrounding ground-truth box $g$, the greater the loss. In the process of backpropagation, $L_{Rep}$ tends to the minimum value, so it can effectively prevent the predicted box is offset the surrounding ground-truth objects and other predicted boxes with different designated objects, respectively.

## 4. Experimental Results and Analysis

The improved algorithm uses the training weight of VGG-16 as the pre-training weight. During the training process, we set the input size of the image to $300 \times 300$, PASCAL VOC 2007 and 2012 training sets are used for training, and then evaluate the algorithm on the PASCAL VOC 2007 test set, the batch size is set to 16, the initial learning rate is set to $4 \times 10^{-3}$. The number of iterations is set to 120 k, and the number of attenuations increases with the number of iterations, for 80 k to 100 k iterations the learning rate is set to $10^{-4}$, and for the last 100 k to 120 k iterations the learning rate is set to $10^{-5}$.

Figure 7 shows the loss of training and the test accuracy. It can be seen from Figure 7a that as the number of iterations increases, the training loss tends to decrease steadily between 20 k and 40 k, and converges around 40 k. Figure 7b shows the test accuracy fluctuates up and down before 40 k iterations and then gradually stabilizes as the learning rate decreases. The final accuracy is 80.7% *mAP*.

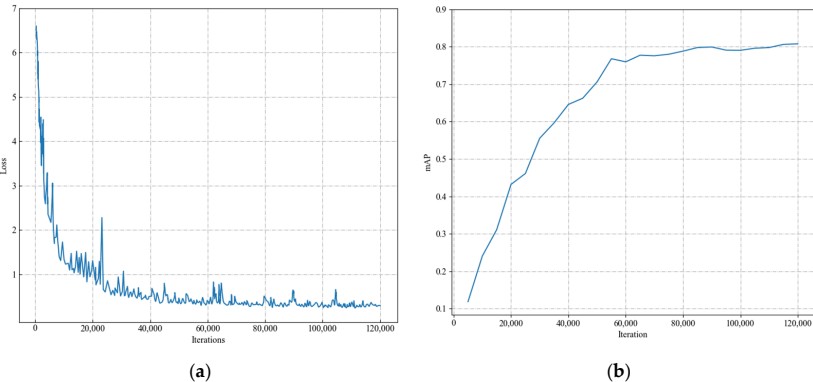

(a)         (b)

**Figure 7.** Training loss and test accuracy of the model. (**a**) Relationship between loss and iteration. (**b**) The *mAP* calculated on VOC2007 test set.

### 4.1. Experiments on PASCAL VOC

We compared the improved algorithm with some other typical object detection algorithms in detection accuracy and detection speed. The methods of comparison are Faster-RCNN, YOLO, SSD, and DSSD. The test results are shown in Table 1. The main considerations are *mAP* and FPS.

**Table 1.** Detection results on PASCAL VOC dataset.

| Method | Backbone | Train | Input Size | *mAP* (%) | FPS |
|---|---|---|---|---|---|
| Faster RCNN [13] | ResNet-101 | VOC 2007 + 2012 | 600 × 1000 | 76.4 | 5 |
| R-FCN [35] | ResNet-101 | VOC 2007 + 2012 | 600 × 1000 | 80.5 | 9 |
| YOLO [14] | GoogleNet-9 | VOC 2007 + 2012 | 448 × 448 | 63.4 | 45 |
| YOLOv2 [19] | Darknet-19 | VOC 2007 + 2012 | 544 × 544 | 78.6 | 40 |
| YOLOv3 [20] | Darknet-53 | VOC 2007 + 2012 | 640 × 640 | 78.3 | - |
| YOLOv4 [36] | Darknet-53 | VOC 2007 + 2012 | 640 × 640 | 84.3 | - |
| YOLOv5_S [37] | Darknet-53 | VOC 2007 + 2012 | 640 × 640 | 81.9 | - |
| SSD300 [15] | VGGNet-16 | VOC 2007 + 2012 | 300 × 300 | 77.2 | 46 |
| DSSD321 [16] | ResNet-101 | VOC 2007 + 2012 | 321 × 321 | 78.6 | 9.5 |
| FSSD300 [18] | VGGNet-16 | VOC 2007 + 2012 | 300 × 300 | 78.8 | - |
| ScratchDet300 [38] | Root-ResNet-34 | VOC 2007 + 2012 | 300 × 300 | 80.4 | 17.8 |
| **OURS** | VGGNet-16 | VOC 2007 + 2012 | 300 × 300 | **80.7** | 26 |

It can be seen from Table 1 that the improved algorithm in this paper has obtained relatively good test results on the VOC dataset. The *mAP* of the SSD algorithm is 77.5%, and our algorithm is 3.2% higher in accuracy than the SSD algorithm, which is improved by 17.3% compared with the YOLO algorithm. Compared with the YOLOv4 algorithm, the accuracy is 3.6% different, and as for the YOLOv5 algorithm, the accuracy is 1.2% different. Compared with the DSSD algorithm, the accuracy is improved by 2.1%, and as for the FSSD algorithm, the accuracy is improved by 1.9%.

### 4.2. Category Accuracy

To verify the accuracy of this method on object detection, we compare 20 types of VOC 2007 with some typical object detection algorithms. As shown in Table 2, we compare the category scores of YOLO, YOLOv2-v5, SSD, DSSD, and our algorithm.

It can be seen from Table 2 that our algorithm not only has an excellent detection effect on normal objects but also has a greater improvement on the detection of small objects, such as, the detection accuracy of aero, boat, bird, and bottle are 89.4%, 72.9%, 81.7%, 65.3%, which are improved by 9.9%, 3.3%, 5.7%, and 14.8% respectively, and it has been comprehensively improved which compare with the traditional SSD algorithm. However, due to the small number of images in some categories of the dataset, the general effect of the detection of categories such as a boat, bottle, chair, and potted plant was not ideal, but the accuracy in these categories of these algorithms is still better than other algorithms. Although there is a certain gap between us and the recent series of YOLOv5 detectors, we will research small object detectors based on the YOLOv5 structure in the future.

**Table 2.** Comparison results of accuracy under different categories.

| Method | SSD | Faster RCNN | DSSD | YOLO | Proposed |
|---|---|---|---|---|---|
| model | VGG16 | ResNet | ResNet | GoogleNet-9 | VGG16 |
| *mAP* (%) | 77.5 | 76.4 | 78.6 | 63.4 | **80.7** |
| aero | 79.5 | 79.8 | 81.9 | 79 | **89.4** |
| bike | 83.9 | 80.7 | 84.9 | 74.3 | **86.7** |
| bird | 76.0 | 76.2 | 80.5 | 62.5 | **81.7** |
| boat | 69.6 | 68.3 | 68.4 | 42.6 | **72.9** |
| bottle | 50.5 | 55.9 | 53.9 | 42.5 | **65.3** |
| bus | 87.0 | 85.1 | 85.6 | 68.3 | **87.2** |
| car | 85.7 | 85.3 | 86.2 | 62.1 | **88.2** |
| cat | 88.1 | 89.8 | 88.9 | 81.4 | **91.1** |
| chair | 60.3 | 56.7 | 61.1 | 42.3 | **62.5** |
| person | 79.4 | 78.4 | 79.7 | 63.5 | **87.7** |
| plant | 52.3 | 41.7 | 51.7 | 41.6 | **55.2** |
| sheep | 77.9 | 78.6 | 78.0 | 65.2 | **78.7** |
| sofa | 79.5 | 79.8 | **80.9** | 54.8 | 78.9 |
| train | 87.6 | 85.3 | 87.2 | 74.9 | **88.2** |
| tv | 76.8 | 72.0 | 79.4 | 62.5 | **79.2** |

*4.3. Comparison of Fusion Methods*

Experiments are performed on the fusion of different shallow feature layers, and the influence of the feature information extracted by the fused feature layer on the detection effect is analyzed. The test results are shown in Table 3. We set the original Conv4_3 layer's feature map as the basic feature map size to compare the fusion effects of Conv3_3, Conv5_3, and Conv6. It can be seen in Table 3, that the fusion of deeper convolutions (Conv5_3 and Conv6) can improve the detection accuracy. Conv5_3 and Conv6 have a larger receptive field, but because the Conv6 Ayer uses dilated convolution and introduces more background noise, the fusion effect with Conv5_3 is more obvious. For the choice of fusion method, concatenate fusion method can reduce the interference of background noise information on the detection results, while the element-sum fusion method can increase the perception of context information. It can be seen from Table 3 that the effect of the concatenate fusion method was better than the elements fusion method.

**Table 3.** Test results of different fusion methods algorithms.

| Layers | Fusion Method | *mAP* (%) |
|---|---|---|
| Conv4_3 | concatenate | 79.3 |
| Conv4_3 + Conv3_3 | concatenate | 79.5 |
| Conv4_3 + Conv6 | concatenate | 80.3 |
| Conv3_3 + Conv4_3 + Conv5_3 | concatenate | 80.4 |
| Conv4_3 + Conv5_3 | concatenate | 80.7 |
| Conv4_3 + Conv5_3 | elementsum | 80.5 |

We compare the visualization of the Conv4_3 layer feature mapping process between the SSD model and the improved model. As shown in Figure 8, the SSD model retains less detailed information about the target feature, and the feature extraction is not very complete. This causes the SSD algorithm to have a weak detection effect on small objects. In contrast, the features extracted by the feature layer that combines Conv4_3 and Conv5_3 can better characterize the characteristics of the object, the semantic information is richer, and the retention of detailed information is more sufficient.

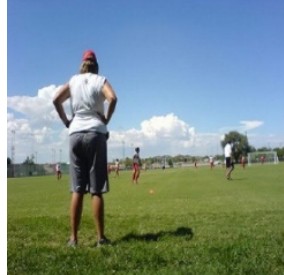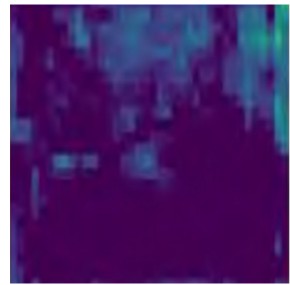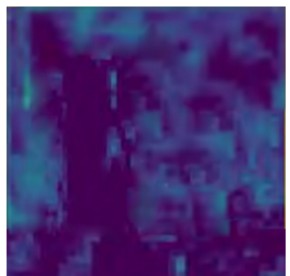

**Figure 8.** Comparison of default box and ground truth frame under different feature maps. From left to right are the original pictures, the conv4_3 feature map visualization of the fusion feature map of conv4_3 and conv5_3.

### 4.4. Experiments on COCO

The COCO dataset includes 80 object categories. We use the COCO 2017 train Val dataset to train our model, use the COCO 2017 test-dev dataset to test different models, and compare the currently popular methods in the object detection field, for example, Faster RCNN, SSD, DSSD, YOLOv2, we use *mAP*@0.5 to evaluate them. As shown in Table 4, our improved SSD algorithm has a more significant improvement than SSD, which is an improvement of 10.6 compared to 41.2 of the SSD algorithm, an improvement of 5.7 of 46.1 of the DSSD algorithm.

**Table 4.** Detection results on COCO dataset.

| Method | Backbone | Train Data | Test Data | *mAP*@0.5 (%) | FPS |
| --- | --- | --- | --- | --- | --- |
| Faster RCNN [13] | ResNet-101-FPN | COCO 2017 trainval | COCO 2017 test-dev | 58.0 | - |
| SSD300 [15] | VGGNet-16 | COCO 2017 trainval | COCO 2017 test-dev | 41.2 | 46 |
| SSD512 [15] | VGGNet-16 | COCO 2017 trainval | COCO 2017 test-dev | 50.4 | 8 |
| DSSD321 [16] | ResNet-101 | COCO 2017 trainval | COCO 2017 test-dev | 46.1 | 12 |
| RetinaNet [39] | ResNet-101 | COCO 2017 trainval | COCO 2017 test-dev | 49.5 | 11.6 |
| **OURS** | VGGNet-16 | COCO 2017 trainval | COCO 2017 test-dev | **50.8** | 23 |

### 4.5. Ablation Study

To further analyze the impact of adding new modules on the detection effect of the algorithm, we conducted ablation experiments on the new modules added to the PASCAL VOC 2007 dataset and studied SSD, SSD with fusion module, SSD with RFB module, and attention mechanism. There are several different combinations of SSD, which were trained on PASCAL VOC 2007 + 2012. Table 5 shows the effect of the modules we added in the experiment, and we found that the method proposed in this paper has a significant improvement.

**Table 5.** Ablation study on the PASCAL VOC 2007 Test Dataset.

| Method | Backbone | Fusion | GC Block | RFB | *mAP* (%) |
| --- | --- | --- | --- | --- | --- |
| SSD | VGG16 | | | | 77.5 |
| SSD | VGG16 | √ | | | 78.6 |
| SSD | VGG16 | | √ | | 80.1 |
| SSD | VGG16 | | | √ | 79.2 |
| SSD | VGG16 | √ | √ | √ | 80.7 |

### 4.6. Experimental Effect Display

To visually show the effect of the improved algorithm, we compare the experimental results of the traditional SSD algorithm and our algorithm. It can be seen from Figure 9, that in each image, objects of the same type are represented by the same color box. It can be seen from this group of pictures that under simple background conditions, both the SSD algorithm and the algorithm in this paper can accurately detect the location and category of the target, but the accuracy of the algorithm in this paper is significantly higher than that of

the SSD algorithm. Under complex background conditions, the algorithm in this paper can detect more targets more accurately. As shown in the first picture, the SSD algorithm can only identify nearby bicycles, and the algorithm in this paper can not only identify bicycles, but also people and cars in the distance are recognized, and the accuracy of bicycles is also higher than that of the SSD algorithm.

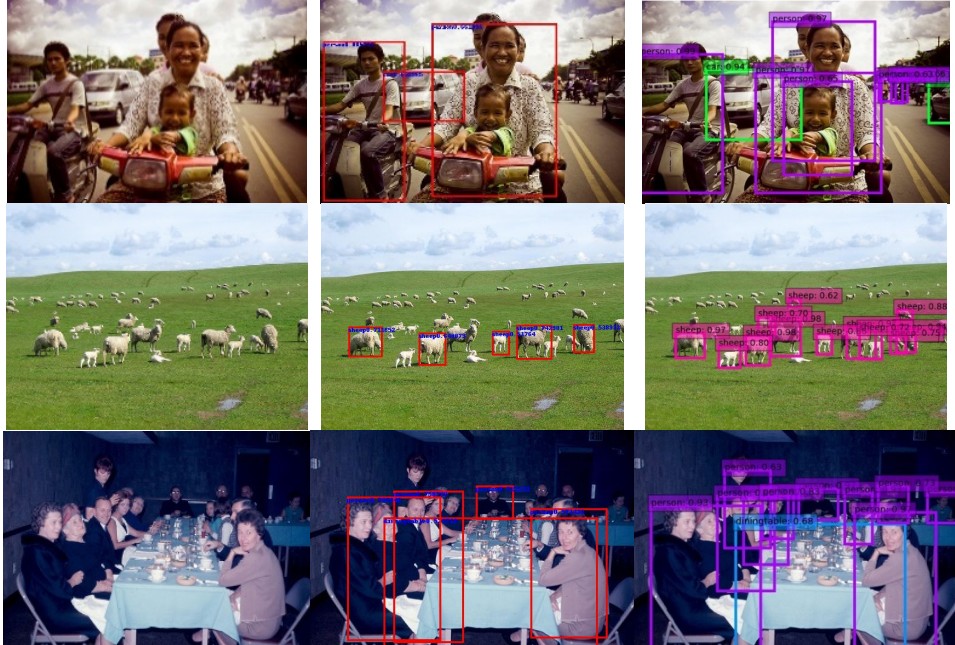

**Figure 9.** Detection examples comparison of the conventional SSD and our proposed algorithm on the PASCAL VOC 2007 test. For each group, the left side is the original image, the middle is the detection result of SSD and the right side is the detection result of our algorithm.

The SSD algorithm has serious false detections and missed detections, and our algorithm improves the situation of false detections and missed detections, which can identify more small objects. Our algorithm has higher accuracy, the overall detection effect is significantly better than the SSD algorithm.

## 5. Conclusions

In this paper, we propose an object detection method based on multi-scale feature fusion and attention mechanism modules. Firstly, we design a feature fusion module that can fuse features of different scales. The fused feature map contains rich details and semantic information. Secondly, we use the GC block as an attention mechanism to obtain global context information. The receptive field module is added to the shallow network to enhance the receptive field of the object and extract the more abundant semantic information of the shallow feature, which is conducted for the detection of small objects. Then we add Repulsion loss to the original loss function, which can partially solve the problem of missed detection. Experimental results display that our model outperforms the traditional SSD model, while the speed is also comparable to other detectors.

In the future, we will continue to improve the detection accuracy of occluded objects in complex backgrounds, and further research the strengthen the information sharing between each feature map of the latest YOLOv5 network. We expect that this improvement in this paper can be applied to the transportation field.

**Author Contributions:** Conceptualization, Z.Q. and T.H.; methodology, Z.Q. and T.H.; validation, Z.Q. and T.H.; formal analysis, T.H.; investigation, T.H. and T.Y.; data curation, T.H. and T.Y.; writing—original draft, T.H. and T.Y.; writing—reviewing and editing, T.H.; writing—original draft, T.Y.; visualization, T.H. and T.Y. All authors have read and agreed to the published version of the manuscript.

**Funding:** This research was funded by the National Natural Science Foundation of China, grant numbers 62176034 and 61905033.

**Informed Consent Statement:** Not applicable.

**Data Availability Statement:** The datasets that support the findings of this study are openly available at PASCAL VOC 2007 and 2012: http://host.robots.ox.ac.uk/pascal/VOC/ (accessed on 2007) and COCO 2017: http://cocodataset.org (accessed on 2017). The authors confirm that the data supporting the findings of this study are available from the corresponding author, [Qu], upon reasonable request, and the data also can get from reference materials.

**Acknowledgments:** The authors wish to thank the associate editors and anonymous reviewers for their valuable comments and suggestions on this paper.

**Conflicts of Interest:** The authors declare no conflict of interest.

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
