# Peer review of "MFFAMM: A Small Object Detection with Multi-Scale Feature Fusion and Attention Mechanism Module"

_applsci, doi:10.3390/app12188940_

Round 1

Reviewer 1 Report

This manuscript presented model to improve SSD performances, but the enhncement and controbution of the presented work compared to prior arts can be  more clearly described. I have comments:

1.  The presented work improves the performances compared to previous SSD works-[15]/[16]. However, the authors did not compare their work with [18] (which used feature-fusion SSD). At least, the presented feature-fusion method should be compared with the method used in [18].

2.  Following 2, please also address and compare the authors’ multi-scale feature-fusion (MSFF) scheme with several prior works claiming MSFF methods (even not for SSD applications). For example, “Multi-Scale Feature Fusion: Learning Better Semantic Segmentation for Road Pothole Detection (arXiv)”; “Multi-Scale Feature Fusion Convolutional Neural Network for Indoor Small Target Detection (Frontiers in Neurorobotics) ”.

3.   Although the presented work improves the performances compared to prior SSD models (Tables 1, 2), it is still worse than YOLO V4/V5. The authors should discuss the evaluation (or design trade-off) of their works with YOLO V4/V5, though they have claimed their plan for YOLO model for the future work.

4.    Also for YOLO, In 4.1, there were no any mentions for YOLO V4/V5, which shows better mAP in Table 1. On the other hand, in 4.2, the statement for category scores of YOLO were mentioned but no data were shown in Table 2.

5.  Minor comments: I suggest that the full name for SSD should be described in the beginning. The texts in Fig.1 are not clear.

Author Response

Dear Editors and Reviewers,

Thank you very much for your thorough review of our paper. We appreciate the time and effort that editors and the reviewers dedicated to providing insightful and valuable comments on our paper.

According to your suggestions, we have made the following revision.

We thank the editors and reviewers for reviewing the previous version of the paper. Their suggestions have enabled us to improve our work. Based on the instructions provided in your letter, we’ve uploaded the file of the revised paper.

We have made point-to-point corrections based on the reviewer’s opinion in the version of our paper.

In the subsequent work, other partners provided contributions, so we added two authors, Le-yuan Gao and Sheng-ye Wang respectively, and we made a detailed explanation in the author contribution statement too.

Once again, thank you very much for your comments and suggestions, and we hope that the revision will meet with your approval.

Thank you and best regards.

Yours sincerely,

Zhong Qu,

E-mail: quzhong@cqupt.edu.cn

Reviewer 2 Report

1. Please increas resolution of a figure.7

2. From table 1., your result obtained 80.7% with 300x300 input size that is better than the method of R-FCN[32] that used higher resolution image. Please explain how you obtained the better result with lower resolution input image.

3. Is the results having dependency on the characteristics of target objects?

4. Is the results having dependency on dataset?

Author Response

(The authors gave the same response as above.)

Round 2

Reviewer 1 Report

I thought the authors have addressed mu comments.